# The Physiological Impact of Melatonin, Its Effect on the Course of Diseases and Their Therapy and the Effect of Magnetic Fields on Melatonin Secretion—Potential Common Pathways of Influence

**DOI:** 10.3390/biom14080929

**Published:** 2024-07-31

**Authors:** Marta Woldańska-Okońska, Kamil Koszela

**Affiliations:** 1Department of Internal Medicine, Rehabilitation and Physical Medicine, Medical University of Lodz, 90-419 Lodz, Poland; 2Department of Neuroorthopedics and Neurology Clinic and Polyclinic, National Institute of Geriatrics, Rheumatology and Rehabilitation, 02-637 Warsaw, Poland

**Keywords:** melatonin action, melatonin hormone, magnetic fields, health influence

## Abstract

Melatonin is a relic, due to its millions-of-years-old presence in chemical reactions, found in evolutionarily diverse organisms. It has a multidirectional biological function. It controls diurnal rhythms, redox homeostasis, intestinal motor functions, mitochondrial biogenesis and fetal development and has antioxidant effects. It also has analgesic and therapeutic effects. The purpose of this paper is to describe the role of melatonin in vital processes occurring in interaction with the environment, with particular reference to various magnetic fields ubiquitous in the life of animate matter, especially radio frequency/extra low frequency (RF/ELF EMF) and static magnetic fields. The most important part of this article is to describe the potential effects of magnetic fields on melatonin secretion and the resulting possible health effects. Melatonin in some cases positively amplifies the electromagnetic signal, intensifying health effects, such as neurogenesis, analgesic effects or lowering blood pressure. In other cases, it is a stimulus that inhibits the processes of destruction and aggravation of lesions. Sometimes, however, in contrast to the beneficial effects of electromagnetic fields in therapy, they intensify pathogenic effects, as in multiple sclerosis by intensifying the inflammatory process.

## 1. Introduction

Numerous scientific papers have reported the widespread presence of melatonin not only in the animal world but also in the plant kingdom. Melatonin is a very old relic compound and is also found in evolutionarily distant organisms such as animals, plants and even bacteria. Although the function of melatonin in plants is not well understood, it can be hypothesized that it is a regulator of plant growth and development that promotes vegetative development in a similar way to indolyl-3-acetic acid (IAA). Melatonin acts as a chemical mediator of darkness, coordinating responses to circadian and photoperiodic environmental cues. It is also a potent antioxidant that stabilizes cells and protects them from environmental damage. These activities serve as the primary regulators of the circadian cycle and many other processes in humans and animals [1]. Incretes such as DHEA (dehydroepiandrosterone), pregnenolone, testosterone, estrogen, progesterone, thyroid hormones, growth hormone and melatonin are sometimes termed superhormones in humans because they not only have a specific physiological role but also have an impact on the prevention and treatment of many diseases [2].

Melatonin as a “superhormone” has multidirectional biological functions and controls diurnal rhythms, mitochondrial biogenesis, redox homeostasis, intestinal motor function and fetal development, among others. Melatonin, as well as its metabolites, has antioxidant activity, making it an endogenous, universal protective factor against many disorders associated with oxidative stress [3].

Melatonin is a small indole that is millions of years old. Both the basic molecule and more complex compounds containing the indole group are commonly found in the tissues of living organisms, both animals and plants. The indole group also occurs as the basis of many substances found in the human body, such as the amino acid tryptophan or the biogenic amine serotonin. The name “indole” comes from the name of the pigment indigo, which contains two indole groups. It is a compound with a very unpleasant odor. It is found in feces and in low concentrations in perfumes [4].

Melatonin’s numerous actions may be receptor-dependent or belong to receptor-independent activities. Receptor-mediated functions encompass diurnal rhythm regulation and tumor suppression. Melatonin’s receptor-independent activities refer to its ability to act in free radical detoxification, thus protecting essential molecules from the damaging effects of oxidative stress caused by ischemia/reperfusion injury (stroke, myocardial infarction), ionizing radiation, drugs, etc. Melatonin has many applications in physiology and medicine [4]. Melatonin that is formed outside the pineal gland is poorly released, while melatonin produced in the pineal gland represents most of the circulating melatonin and the fraction of melatonin that enters the brain, specifically the third ventricle, through the pineal gland’s cistern. Moreover, melatonin acts also as a paracrine and autocrine factor [4,5]. Thus, it appears that tissue processes are impacted by both locally produced and circulating melatonin.

The purpose of this study is to describe the role of melatonin in its role concerning vital processes in interaction with the environment, including with particular reference to the various magnetic fields ubiquitous in the life of animate matter, especially radiofrequency/extra low frequency magnetic fields (RF/ELF EMF).

## 2. The Physiological Role of Melatonin

When in 1958, Aaron B. Lerner and his colleagues reported that they had isolated and characterized a serotonin derivative from bovine pineal gland, they were unable at the time to predict how many functions this molecule might ultimately perform. On the other hand, as dermatologists, they knew that pineal gland extracts contained an active factor that affected cutaneous chromatophores, brightening amphibian skin. This is why the name they chose for the newly discovered molecule is partially related to its effect on skin pigmentation (“mela” from melanin and “tonin” from serotonin). As a result of this discovery, the pineal gland came to be regarded as an active endocrine gland. Biochemists have identified the melatonin synthesis pathway, and physiologists have described the role of the pineal gland in the regulation of seasonal reproduction in photoperiod-sensitive mammals and its requirements for sympathetic innervation to function [4].

During the day, melatonin is produced in small amounts (its serum concentration ranges from 5 to 20 pg/mL), while it is mainly produced at night to reach serum concentrations several times higher than during the day (80–120 pg/mL) during the peak secretion period (between midnight and 3 a.m.). Melatonin—previously thought to be produced only by the pineal gland—is synthesized in various other tissues, such as the immune cells, brain, retina, heart, blood vessels and gastrointestinal system, with the latter probably being the main source of melatonin, as the estimated production level is 400 times greater than in the pineal gland. It is of importance that melatonin synthesized outside the pineal tissue mainly acts locally. In the gastrointestinal tract, it is responsible for slowing digestive processes, although it stimulates the secretion of enzymes by the pancreas [6].

Melatonin acts in multiple ways, and given that it is millions of years old, it may have improved its functions. For a long period after its discovery, it was presumed that melatonin mediated all of its actions by binding to a limited number of cells, such as neurons of the suprachiasmatic nucleus (SCN). Scientists identified two relevant membrane receptors that are currently called MT1 (Mel1a) and MT2 (Mel1b) receptors. Both belong to the superfamily of G-protein-coupled trans-membrane receptors. Additionally, there is also an orphan molecule known as the melatonin-related receptor (MRR; also called GPR50). It shares 45% homology with MT1 and MT2, but its function is mostly unknown [4,7]. Some polymorphisms of this gene in women are associated with an increased risk of developing bipolar disorder, major depression and schizophrenia [7].

Another polymorphism of the GPR50 gene has been linked to higher levels of fasting triglycerides and lower levels of circulating high-density lipoprotein. Contrary to the initial belief that membrane melatonin receptors could only be located on a few cell types, later studies have actually found them on many other cells, and they could potentially be present on the membranes of all cells [8].

There is also a so-called MT3 receptor, an additional subtype of melatonin receptor, that has been detected in amphibians and birds and is localized in the cytosol of some cells. This receptor has a low affinity for iodomelatonin and is not coupled to a G protein; it was identified as quinone reductase 2, a detoxification enzyme. Moreover, melatonin can perform some of its actions by interacting with nuclear receptors known as retinoid Z receptors (RZRs) or retinoid orphan receptors (RORs). Nuclear receptors can be variously distributed in tissues, and their function is probably best described in the immune system. Interestingly, in addition to its receptor-mediated action, melatonin has the ability to freely penetrate the cell membrane barrier, being highly lipophilic and thus effectively protecting the cellular compartment from oxidative damage [4,6].

As mentioned, melatonin exerts receptor-dependent as well as receptor-independent effects. It binds to two membrane receptors, MT1 and MT2, and through several pathways influences a number of physiological reactions. In some cases, MT1 and MT2 undergo homo- and/or heterodimerization and can interact with some nuclear receptors. The topic of the existence and function of orphan receptors, ROR and RZR, has been widely discussed in the literature. The cytosolic receptor, MT3, is quinone reductase 2 [3].

The receptor-independent actions of melatonin (and its metabolites) are exerted thanks to its ability to scavenge reactive nitrogen species (RNS) and reactive oxygen species (ROS). It creates a protection against a wide range of toxins and processes that produce highly harmful compounds. The diagram lists only some of the identified physiological and molecular functions of melatonin (Table 1) [4,5,6,7,8,9,10,11,12].

Melatonin, acting through a variety of pathways, positively affects human health and has a good safety profile. Additionally, according to recent animal studies of developmental programming, melatonin use in early life may prevent hypertension in later years. In a prenatal model of dexamethasone exposure, melatonin therapy in the mother prevented hypertension in the offspring, accompanied by the reversion of the reduction in the number of nephrons caused by the glucocorticoid. Melatonin plays a significant role in the prevention and control of hypertension. With a better comprehension of the early causes of hypertension and a substantial increase in the early use of melatonin, it is expected that melatonin will be used as a reprogramming therapy to reduce the overall burden of hypertension [3].

Due to the heterogeneous effects of melatonin, the mechanisms of reprogramming in melatonin therapy in early life may be interrelated and complex, making it difficult to ascertain their relative importance. Further research needs to adopt a unified approach to determine the different mechanisms of action in early melatonin use in just one experiment [3].

Melatonin’s ability to prevent free radical-induced molecular damage and cellular injury manifests itself during direct exposure of cells to damaging stimuli, which include smoking; the ingestion of toxins, heavy metals, alcohol and prescription drugs; ultraviolet and ionizing radiation; ischemia/reperfusion injury (during myocardial infarction or stroke); neuro-degenerative diseases; and severe inflammation [3,4,6].

Moreover, it appears that one of the abovementioned nuclear receptors, transcription factor RZR(RORβ)/ROR alpha, mediates direct gene regulatory effects by melatonin, as it was thought before. For some time, some studies have shown that melatonin indirectly regulates the transcriptional activity of RORs through intermediate steps, i.e., ligand-dependent transcription factor that modulates the transcription of target genes by binding to ROR response elements in target genes. Some studies have shown that ROR receptors mediate melatonin’s effects and also indicate that melatonin can regulate the transcriptional activity of RORs by increasing the transcription of ROR target genes. However, RORs have also been found to act independently of melatonin-induced signaling. Orphan ROR receptors influence physiological processes such as diurnal rhythm, oxidative stress, the modulation of inflammation, cerebellar and lymphoid tissue development, retinal development, bone formation and lipid metabolism. It appears that ROR may be a point of resolution in the treatment of autoimmune diseases and cancer as well as metabolic diseases, including obesity, diabetes and other diseases. Determining the action of new natural and synthetic ROR ligands is therefore an important cognitive aspect. Further research is needed to establish targeted therapies for the listed processes and diseases [9].

## 3. Influence of Melatonin on the Course of Diseases and Their Therapy

The analgesic effect of melatonin was confirmed by Ahmad et al. [13]. Melatonin substitution may include situations involving the extensive use of melatonin in the treatment of various gastrointestinal diseases, inflammation, cancer, mood disorders and others. Low-quality evidence (downgraded for imprecision) presented by Andersson et al. [14] indicates that, compared to placebo, 4–8-week melatonin treatment reduces pain intensity.

As mentioned above, the majority of melatonin’s activities are receptor-dependent; its molecule-dependent activities, although less numerous, include such prominent examples as free radical scavenging. Melatonin, as a potent antioxidant, is directly responsible for free radical scavenging, as are its metabolites. The free radical scavenging as well as anti-inflammatory effects have a substantial impact on inflammation and pain etiology [14].

Smolensky, M.H., et al. [15] suggest that ALAN (Artificial Light at Night) causes deleterious effects that may include insufficient, abnormally timed exposure or the near complete absence of natural daytime sunlight exposure and associated temporal signals, including those carried by active vitamin D synthesis metabolites exhibiting diurnal and annual patterns. This hypothesis requires extensive research and verification. Exposure to today’s abnormal artificial light environment, devoid of daytime UV-B and other biologically active wavelengths, rich in ALAN, resulting in vitamin D deficiency and melatonin suppression, is often coupled with CTS (circadian time structure) disturbances, and begins in early life. Consider here the previous and continuing exposure to artificial day/night light, as well as regular behaviors, such as taking certain classes of drugs that are α1 and β1 receptor antagonists, which imitate the effects of ALAN on melatonin suppression and probably induce CTS disorders.

These mechanisms are responsible for the increased incidence of hormone-dependent cancers and perhaps other conditions such as mental, cardiovascular and diabetes, the so-called diseases of civilization [3,4,6,7,13,15]. The broad knowledge of these issues will help prevent them. The potentially harmful consequences of the combination of inhibition of vitamin D and melatonin synthesis as a result of living under artificial light have never been comprehensively investigated in chronobiological or epidemiological studies and should have been investigated by now. Every effort should be made to comprehensively assess the effects and potential negative consequences of a contemporary life in an environment characterized by too little sunlight during the day combined with too much artificial light at night [15].

A comparative analysis [16] demonstrated that long-lasting controlled-release melatonin formulation, unlike immediate-release formulation, lowered sleep systolic blood pressure by 3.57 mm Hg. Furthermore, diastolic blood pressure during both sleep and wakefulness was also reduced. Melatonin’s mechanisms of action include inducing vasodilation by directly blocking calcium channels and increasing the production of cyclic guanosine monophosphate and nitric oxide in the endothelium, inhibiting the sympathetic nervous system and reducing norepinephrine production, exhibiting its antioxidant activity, and activating the parasympathetic nervous system. Moreover, as it is usually taken before bedtime, melatonin may be used in a potential therapy for sleep hypertension, and can increase the duration and quality of sleep, reducing blood pressure during sleep. The low secretion of endogenous melatonin during sleep is related to sleep hypertension, lack of blood pressure drop and cardiovascular diseases.

As it was described by Huang, X., et al. [17], when melatonin was administered to regulate sleep, serum BDNF (brain-derived neurotrophic factor) levels increased, indicating a relationship between melatonin levels and BDNF levels. It was also suggested that BDNF exerts a therapeutic effect on CRSWD (circadian rhythm sleep–wake disorders). Accordingly, this study may provide a theoretical basis for treating CRSWD with melatonin.

In the case of multiple sclerosis (MS), abnormal melatonin synthesis has been observed in connection with the different lifestyles of patients, with some of them exhibiting sometimes high levels of this hormone. A study in an experimental mouse model of multiple sclerosis showed that darkness and the administration of exogenous melatonin increased the inflammatory process with the infiltration of monocytes and reduced oligodendrocyte neurogenesis. These data show that it is necessary to monitor diurnal changes in melatonin level in every patient with multiple sclerosis, taking into account diet and lifestyle, to avoid overdosing. Melatonin is not a panacea and, contrary to previous belief, it does not have universal positive regulatory effects and can be overdosed [18].

An interesting observation is made by authors Vuković et al. [19], who suggest that women with multiple sclerosis have smaller pineal glands, which can theoretically be explained by the lack of input stimuli and the resulting reduction in gland volume. In addition, the risk of multiple sclerosis is reduced with larger pineal volumes. Both of these phenomena may have a role in the pathogenesis of multiple sclerosis.

Melatonin is regarded as a component of the innate defense system and a very effective protector of the pancreas. In animal model studies, it has been demonstrated that pretreatment with melatonin can be used to prevent pancreatitis and dramatically decrease pancreatic tissue damage in rats subjected to acute pancreatitis [6].

MEL in vivo effectively reduced the expression of the HIF-1α/GLUT1/NLRP3 pathway in the lung tissue of LPS (lipopolysaccharide)-treated mice. MEL regulates the activation of the ROS/HIF-1α/GLUT1/NLRP3 pathway in alveolar macrophages via the MT1 receptor. In the study presented here, it significantly attenuated LPS-induced lung injury by inhibiting ferritinophagy, further alleviating ARDS (acute respiratory distress syndrome) induced sepsis [10].

Reduced melatonin levels and the functional correlation between melatonin and insulin play a role in the pathogenesis of type 2 diabetes (T2D). In addition, genomic studies have shown that rare melatonin 1b MT1b receptor (MTNR1B) variants are also correlated with impaired glucose tolerance and increased risk of T2D. Moreover, treatment with exogenous melatonin in studies in animal cell models and patients with diabetes has shown a strong effect on alleviating diabetes and other correlated complications of the disease. This defines to some extent the role of melatonin in glucose homeostasis. However, there are also conflicting reports on the effects of melatonin supplementation [20].

Due to its antioxidant properties and mitochondrial function, melatonin was used in COVID-19 infection. This study [21] evaluated whether melatonin treatment compensated for the altered redox homeostasis in the serum of patients with COVID-19. The treatment promoted an increase in antioxidant enzyme activity, which contributed to an increase in TAC (total antioxidant capacity) and restored redox homeostasis. The aforementioned hormone also modulated glucose homeostasis, acting as a glycolytic agent and inhibiting the Warburg effect. The findings of Alomari et al. [22] underscore the evidence supporting the antiviral properties of melatonin. In silico studies have identified melatonin as a candidate against COVID-19, reducing cytokine storm-related respiratory responses.

The effect of melatonin on immunity is very complex and involves several types of immunocompetent cells. This is evidenced by a number of papers [10,12]. Melatonin inhibits influenza virus infection and improves lung function and lung damage in AECOPD by inhibiting IL-1β/STAT1-induced M1 macrophage polarization and apoptosis in an MTs-dependent manner. Melatonin can be considered a potential therapeutic agent in AECOPD (acute exacerbations of chronic obstructive pulmonary disease) caused by influenza virus infection [12].

Another approach is to use melatonin and macrophages to potentially accelerate bone healing. The microsphere, containing melatonin (MT) and calcium phosphate (CaP) crystals, is injectable and degradable, and the substances it releases cause M2 macrophages to polarize, subsequently reprogramming them and increasing osteogenesis [11].

Summarizing the role of the hormone, melatonin is correlated with memory, exerting a direct effect on hippocampal neurons. It also controls posture and body balance. Melatonin has antinociceptive, antidepressant, anti-anxiety and regulatory effects on the locomotor system. Melatonin has neuroprotective, blood pressure-lowering, pain-modulating, differentiation of vascular retina, osteoblast, seasonal reproduction, ovarian physiology, anticancer, antiviral and antioxidant properties. Gonadotropin-releasing hormone (GnRH) secretion from hypothalamic neurons is regulated by melatonin, which additionally affects the synthesis of folliculotropic hormone (FSH) and luteinizing hormone (LH). In granulosa cells, progesterone production is promoted by melatonin. Melatonin also inhibits estrogen receptor expression and estrogen activation. Melatonin has been shown to be very helpful in the treatment of neurological disorders such as Parkinsonism, Alzheimer’s disease, cerebral edema and traumatic brain injury, depression, cerebral ischemia, hyperhomocysteinuria and phenylketonuria and has a complementary effect in the treatment of glioma. Melatonin has been shown to inhibit amyloidosis [9,13,23].

## 4. Influence of Magnetic Fields on Living Organisms

In the case of environmental and industrial magnetic fields, one of the side effects of any electrical device is the electromagnetic field generated in its surroundings during operation, which affects the health of living beings in its vicinity. All organisms, including humans, are exposed every day to a multitude of environmental fields characterized by different physical parameters. Therefore, it is important to accurately determine the effects of electromagnetic fields on physiological and pathological processes occurring in cells, tissues and organs. Numerous epidemiological and experimental data suggest that extremely low-frequency magnetic fields generated by power lines and electricity-powered devices, as well as high-frequency electromagnetic radiation emitted by electronic devices, have a potentially negative impact on the circadian cycle. According to the European Commission, non-ionizing radiation can be divided into several levels [23]:-Static fields,-Extremely low frequency fields (ELF fields),-Intermediate frequency fields (IF fields),-Radiofrequency fields (RF fields).

Organisms living on Earth are exposed to the geomagnetic field (GMF, current intensity of 25–65 μT), and numerous species use it for long-distance navigation and orientation. GMF affects organisms mostly by protecting them from cosmic radiation, including solar wind, and by preventing oxygen from escaping into outer space, making the Earth more hospitable. However, it is also possible for organisms, including humans, to be exposed to hypomagnetic fields (HMF, static magnetic field of <5 μT), for example, during long-duration space flights or in some artificial environments, including magnetically shielded rooms, as well as in areas of the globe where magnetic radiation is critically low. Exposure to HMF can induce behaviors resembling central nervous system (CNS) dysfunction, such as nervousness and decreased analgesia induced by stress in adult mice, amnesia in Drosophila flies and chickens and cognitive impairment in humans [24].

A deficiency of magnetic fields can adversely affect biological processes, especially when the radiation of the earth’s magnetic field is reduced. Shielding of the earth’s magnetic field can adversely affect neurogenesis, and this effect is closely correlated with ROS concentrations. Recently, it was observed in mice that exposure to a HMF can negatively affect neurogenesis in adult hippocampus and hippocampus-related cognitive functions. The same study demonstrated the role of ROS in the effects of the hypomagnetic field, although the mechanistic reasons for this effect are not clear. Another study [25] described flavin-superoxide radical pair-based mechanism to explain ROS production modulation and adult hippocampal neurogenesis impairment in a hypomagnetic field. The computational results favor the singlet radical pair over the triplet one. The described model predicts that the effect of the hypomagnetic field on triplet/singlet output has strength comparable to the strength of the effects observed in experimental studies on neurogenesis in adult hippocampus [18]. The restoration of the geomagnetic field (GMF) reverses the deleterious decrease in ROS forms. GMF plays an important role in neurogenesis in adult hippocampus by maintaining adequate levels of endogenous ROS [23] Maintenance of adequate ROS levels by GMF is required for normal adult hippocampal neurogenesis and function [25].

Due to its many beneficial effects, very low-frequency MF exposure is used in the physiotherapy of certain neurological diseases and musculoskeletal overload syndromes. Therefore, it is important that medical devices do not cause disturbances in the circadian cycle. Such disturbances occur in the circadian cycle in children with attention deficit hyperactivity disorder (ADHD), in people with Parkinson’s disease or Alzheimer’s disease, and in shift workers [26,27].

Bodrova, R.A., et al. [28] conducted a study of 200 patients with COVID-19 pneumonia using physical therapy to complement standard treatment. The study group of 100 pa- patients, received magnetic therapy with a maximum induction of 25 mT, a frequency of 1–100 Hz, increasing daily by 10–20 min in a cycle of 15 treatments. The comparison group contained 100 patients who underwent placebo magnetic therapy. Low-frequency magnetic therapy included in the comprehensive rehabilitation of patients after COVID-19-associated pneumonia caused improved general well-being, improved patients’ respiratory function, reversed residual infiltrative lesions in the lungs, alleviated inflammatory process symptoms (including in laboratory tests), shortened rehabilitation and disability time, increased exercise tolerance, normalized the psycho-emotional state, and, as a result, restored activities in daily life and improved patients’ quality of life. No side effects were recorded. Similar results were obtained in a study by Jankowska, A., et al. [29].

COVID-19 morbidity is associated with excessive inflammatory response including exaggerated cytokine production in the lungs, leading to acute respiratory distress syndrome. The cellular mechanisms underlying these so-called “cytokine storms” are controlled by the Toll-like receptor 4 (TLR4) and by ROS signaling pathways. Stimulation with magnetic fields (e.g., pulsed electromagnetic fields) is a non-invasive therapy known to have anti-inflammatory effects and regulate ROS signaling pathways. In a study, it was shown [30] that the daily exposure of human cell cultures to electromagnetic fields, specifically to daily 10 min periods of pulsed electromagnetic fields (PEMF) or static low-level magnetic fields, significantly reduced the inflammatory response induced by the TLR4 receptor signaling pathway. Since there are no known adverse effects of current electromagnetic field therapies and the therapies are already approved for some medical applications, protocols have been developed for their verification in clinical trials involving COVID-19 infection. These therapies are inexpensive, simple to implement and can help resolve acute respiratory failure in patients with COVID-19 both at home and in the hospital [29,30].

Another application of magnetic fields took place for patients with long COVID-19. Over the course of 5 weeks, a patient with post-COVID-19 fatigue syndrome was treated with 10 sessions of pulsed high-density magnetic flux electromagnetic field treatment. These sessions were applied after 5 months of unsuccessful kinesiotherapy. Symptoms such as general fatigue, ability to work, psychological well-being and quality of life improved markedly already during the treatment and were stable after 6 weeks. The application of pulsed electromagnetic field therapy may be a physical method in the treatment of fatigue syndrome after COVID-19, which may reduce clinical and economic health consequences [31].

According to Markov, M. [32], magnetic and electromagnetic fields (PEM) can be used to treat various medical problems, including pain, locomotor system injuries and vascular and endocrine disorders.

Recent reports aim to provide a closer look at MF magnetic field therapy in brain tumors. With significant advances in diverse physical therapies, magnetic field therapy, a potential complementary therapy, has become generally known for its beneficial effects, such as its painlessness, invasiveness and reusability. As MF is able to selectively kill cancerous cells by affecting the cell cycle, the prospects for treating intracranial cancers such as gliomas are promising. Several investigators have studied ELF-PEMF mechanisms that affect glioma cell lines, calcium ions, autophagy and apoptosis and have suggested that ELF-PEMF probably enhances the effects of chemotherapy and RT [33].

PEMF’s mechanisms of action can lead to the activation of specific signaling pathways, including the ability to activate nitric oxide, as well as moderate radical reactions that can ultimately lead to, for example, reactions similar to those including NFκB (which is a protein complex playing a crucial role in regulating the immune response to infections). In the right way, these reactions can cause a type of cell protection and ultimately lead to the suppression of inflammatory signals such as the interleukins IL-1β and TNF-α. In a study by Funk, R.H., the pulse duration was 1.3 ms, the frequency was 75 Hz, the cycle filling was 1/10 and the magnetic field intensity was 2.3 mT [32].

The phenomena occurring during the application of magnetic fields coincide with melatonin activation pathways, and reactions related to the action in calcium channels and calmodulin-related processes, also include Ca^2+^/calmodulin-dependent nitric oxide synthases, such as (neuronal) nNOS, (endothelial) eNOS and Inducible NOS (iNOS) and singlet-triplet conversion in radical pair reactions, leading to an increase in triplet radical pair concentrations, ultimately increasing oxidative stress. For example, a weak RF magnetic field affects the formation of reactive oxygen species (ROS) from hydrogen peroxide (H_2_O_2_) and superoxide (O^2−^). Normally, melatonin causes opposite reactions to RF/ELF-EMF, hence its protective effect. Nevertheless, magnetic field induction that is too low causes the inhibition of neurogenesis [9,25,34].

## 5. Potential Effects of Magnetic Fields on Melatonin Secretion and Subsequent Possible Health Effects

Today, the populations of Europe and North America are widely exposed to magnetic fields of 50 Hz and 60 Hz, respectively, which are generated by commonly used electrical devices. In several studies that were conducted on power plant workers exposed regularly to 60 Hz magnetic fields, a reduced urinary excretion of 6-sulfatoxymelatonin has been noticed [35] with significant changes being observed after the second day of the work week. Moreover, this effect was most noticeable in those with low light exposure in the workplace. The results were slightly different from the study by Juutilainen et al., where a greater reduction in the concentration of melatonin was observed in those with greater exposure to light [36]. Nonetheless, it is evident that environmental and network fields disturb nocturnal secretion of melatonin, which has negative effects on health [23]. Some previous studies do not fully confirm these observations [35,37].

In a study by Woldańska-Okońska, M., et al. [38], it was observed that in both magnetostimulation programs, one with M2P2 cyclotron resonance and the other without M1P1, that the duration of melatonin secretion increased in the morning one month after application, which may have implications for the condition of patients with SAD. In this case, there was no nocturnal decrease in melatonin secretion. The effect of the magnetic field persisted one month after the application, which supports the hypothesis that biological hysteresis phenomenon occurs after magnetic stimulation. From the graphs of the circadian cycle of melatonin after magnetic stimulation in M1P1 and M2P2 programs [38], as well as magnetotherapy [39], it is evident that melatonin secretion is affected by magnetic field parameters, the application system and the time and length of application. The influence of parameters such as polarity type and vector should be considered. It appears that magnetic stimulation has unifying properties that regulate melatonin secretion, but this needs further study.

Interestingly, the rise in serotonin concentration observed in the M2P2 program (with cyclotron resonance) was not reflected in the circadian melatonin curve [40], which could be expected in the context of serotonin to melatonin conversion. It is probable that the effect observed in the conversion is not dependent on the stimulation of serotonin N-acetyltransferase or 5-hydroxyindole-O-methyltransferase. In M2P2 program, magnetic stimulation affected serotonin synthesis and did not change the intensity of the conversion to melatonin.

When using magnetic fields as a physical treatment for degenerative brain diseases, it is important not to disturb natural circadian cycle as it could deprive patients of the positive effects of melatonin and not to induce cortisol secretion outside the normal diurnal rhythm [41,42].

Evidence of melatonin’s analgesic effect is presented in quite a few papers [13,14] and suggests that melatonin treatment reduces pain intensity compared to placebo. Situations where melatonin may be used include the treatment of many gastrointestinal diseases, inflammation, cancer and mood disorders, among others. Magnetic fields can also affect the sensation of pain [43]. As was mentioned above, a prominent example of molecule-mediated activity of melatonin is free radical scavenging. This activity in addition to the anti-inflammation effects of melatonin have a substantial impact on inflammation and the etiology of pain [13,43]. Too low a level of magnetic field induction reduces free radical activity, and thus melatonin secretion (hypomagnetic field) decreases. Unchanged melatonin levels after magnetic field application result in the increased activity of pathways that follow a similar course when both magnetic fields and melatonin are applied. This may include responses to endorphins, nitric oxide, inflammatory mediators or neuromuscular reactivity [44].

Evidence on melatonin being an antioxidant and potential drug indicates that this molecule is not just a sleep hormone produced by the pineal gland. Numerous studies have demonstrated its ability to combat oxidative stress as it can scavenge free radicals directly or by inducing the activation of antioxidant enzymes. While considered safe, non-ionizing electromagnetic fields cause biological changes in the body, e.g., increase lipid peroxidation and create a state of excessive oxidative stress. The studies on electromagnetic fields on melatonin in humans are not conclusive, as some of them provided conflicting evidence, while others presented only negative results. Nonetheless, as it is possible that electromagnetic fields affect the usefulness of melatonin to control external stressors, researchers have investigated the radio-protective role of melatonin. The free radical scavenging action of melatonin can be enhanced by reducing the conversion factor of singlet-triplet radical pairs and the concentration of triplet products. Moreover, it is worth highlighting the potential therapeutic benefits of melatonin as an agent that nullifies the harmful effects of electromagnetic fields in general and the occurrence of electromagnetic hypersensitivity (EHS) in particular [45].

It is not certain how magnetic fields (MF) influence melatonin production in humans, as evidence is still somewhat limited and not always consistent. Differences in results can be partially explained by studies suggesting that light can have an impact on the pineal gland’s response to MF. On the other hand, it is now known that by altering magnetic field parameters it is possible to change the degree to which melatonin secretion decreases at night [36].

Epidemiological studies have shown an elevated risk of leukemia in both adults and children living close to overhead high-voltage power lines, even beyond the range of the electric and magnetic fields created by these lines. The coronal ions generated by these fields form a plume that is transported by the wind for distances of up to several hundred meters. As a result, highly variable disturbances in the atmospheric electric field of up to hundreds of V/m are generated for periods of seconds or minutes. They supposedly interfere with nocturnal melatonin synthesis and associated diurnal cycles, possibly causing adverse health effects, including hematopoietic neoplasms in humans [46].

A broad issue is the impact of magnetic fields disrupting nighttime melatonin secretion on neurogenesis. The melatonin-induced stimulation of NSCs or NSPCs (neural stem/progenitor cells) to proliferate and/or differentiate into neurons has been extensively documented (Table 1). However, melatonin has also occasionally been reported to reduce NSC proliferation, e.g., during a normal light/dark cycle, where cell division in scotophase was found to be reduced. Moreover, it has been repeatedly reported that melatonin promotes the survival of NSCs/NSPCs under various conditions. As it affects survival and the proliferation and differentiation of NSPCs, melatonin has great medicinal value, and it could be used in the treatment of neurodegenerative diseases and various brain and spinal injuries. Problems often encountered after the transfer of NSPCs to dysfunctional or damaged sites include poor survival and abnormal differentiation. With the use of melatonin, it was possible to overcome these difficulties and produce significant improvements, at least at the preclinical level. In the case of unfavorable magnetic fields, the unfavorable effect exerted on NSCs or NSPCs (including through melatonin suppression) disrupt the initiation of the brain plasticity process after brain damage [5]. Various parameters of magnetic waves used in physical therapy devices have been tested and indicate a positive effect on brain plasticity after strokes, which indirectly indicates their beneficial effect on neurogenesis [47]. The beneficial effects of magnetic fields are observed in the treatment of multiple sclerosis in terms of analgesic effects, quality of life and cognitive functions [48,49,50].

Due to the primacy of melatonin in the regulation of body processes, as well as the ubiquity of magnetic fields in our lives, including environmental ones, the meeting of both factors must give rise to several effects of these interactions (Table 2). Magnetic fields disturb or not the secretion of melatonin depending on their parameters. The deficiency of magnetic fields causes a number of effects with far-reaching consequences, including affecting the proliferation and transformation of stem cells, including neurogenesis [23]. Melatonin, in turn, regulated by “clock genes”, rhythmizes most circadian rhythms not only in humans but also in animals. The total biological regulator and shield that is melatonin causes the adaptive potential created in space over millions of years that preserves life on Earth [25,30].

The limitations of the work are related to the selection of specific elements concerning the topics described, so that a certain subjectivity cannot be avoided. Studies on the correlation of the mechanisms of action of melatonin and magnetic fields have not been conducted or are not available in the literature, hence the lack of strictly scientific confirmation of these observations. Nevertheless, it seems interesting to the authors to draw attention to a possible aspect of the interaction of these factors.

## 6. Conclusions

Melatonin, as a super-regulator, affects most of the life processes of cells, tissues and organs, synchronizes these processes in the circadian rhythm at the basic level and by regulating other hormones. The pathways through which melatonin exerts its effects on the body are often the same pathways through which magnetic fields act. Melatonin sometimes positively enhances the electromagnetic signal, enhancing health effects such as immune stimulation, neurogenesis or analgesic effects or lowering blood pressure. In other cases, it is a signal that inhibits the processes of destruction and the intensification of disease lesions. However, sometimes, contrary to the beneficial effects of electromagnetic fields in therapy, melatonin intensifies the pathogenic effect, as in multiple sclerosis, by intensifying the inflammatory process. These phenomena are clinically relevant, due to the use of magnetic fields in therapy and in diagnosis, so it is worthwhile to use the knowledge of the synergy of melatonin and magnetic fields in the aforementioned areas.

## Figures and Tables

**Table 1 biomolecules-14-00929-t001:** Melatonin functions in humans and animals (melatonin functions in invertebrates and plants are not included) [3,4,5,7,8,9,10,11,12].

MELATONIN Pineal-Derived and Locally Produced in Many Cells			
Receptor-dependent			Receptor- nondependent
Membrane MT1, MT2, GPR50	Nucleus ROR, RZR (RORβ)	Cytosol MT3, calmodulin	Free radicals scavenging ROS, RNS
Seasonal reproduction Retinal physiology Sleep promotion Circadian modulation Bone growth Blood pressure modulation MT1 vasoconstriction, MT2 vasodilation, varied effects on stem cells Neurogenesis Bipolar disorder, depression Effects on lipoproteins Affects macrophages by inhibiting ferritinophagy and alleviates ARDS Inhibits IL-1β/STAT1-induced M1 macrophage polarization and apoptosis Reprograms macrophages and increasing osteogenesis	Immune modulation Antioxidant enzyme regulation Regulation of genes Gut microbiota inhibits apoptosis of Leydig cells Promotes the secretion of testosterone Suppression of liver and colon cancer Regulation of hair growth	Detoxification Enzyme regulation Suppression of calcium signaling once the signaling cascade is initiated, activating Ca^2+^-ATPases and calcium pumps Cell cycle modulation	Protection against: Ionizing radiation UV radiation Ischemia/reperfusion Heavy metal toxicity Alcohol toxicity Drug toxicity

**Table 2 biomolecules-14-00929-t002:** Some examples of the effects of magnetic fields and melatonin on living organisms (**↑**—improvement; **↓**—worsening). Potential common pathways of influence.

No	Parameters of Magnetic Fields	Research Subject	Melatonin Action and Similar LFMF or Static	Common Metabolic Pathway	References
1.	A 4–8-week melatonin treatment. Magnetostimulation, M3P3 program, magnetotherapy 40 Hz	Patients with pain	Melatonin reduced pain intensity, 3 weeks of use magnetic fields Pain decreasing ↑	Melatonin’s mechanisms of action as well as magnetic fields causes the removal of oxidative and nitrosative radicals, as well as the inhibition of pro-inflammatory cytokines by inhibiting the production of PGE2 and NO	[13,14,32,44]
2.	Magnetic stimulation program M2P2, 15–190 µT	Patient with HA with melatonin supplementation	Hg. diastolic blood pressure was also reduced HA ↑	Melatonin’s mechanisms of action as well as magnetic field actions include vasodilation by blocking calcium channels and increasing the production of cyclic guanosine monophosphate and nitric oxide in the endothelium, exhibiting its antioxidant activity.	[3,16,35]
3.	Melatonin in doses of over 5 mg Magnetostimulation M2P2 program	Multiple sclerosis patients (MS)	Melatonin in great doses reduces immunity ↓ Magnetostimulation stimulates immunity ↑ Analgesic and cognitive functions—both factors increase ↑	Impact on immunity -opposite action Analgesic effects cognitive functions and quality of life synergism	[18,48,49,50]
4.	Exogenous melatonin in studies in animal cell models and patients with diabetes LFMF used in physiotherapy	Animals and humans with T2D	Reduced melatonin correlate with impaired glucose tolerance and increased risk of T2D; strong effect on alleviating diabetes and other correlated complications of the disease Diabetes treatment ↑	Effects sugar levels and the course of diabetes, reduces the symptoms of diabetic neuropathy	[20,32]
5.	Melatonin was used in infection Magnetic fields used in infection MF pulse duration was 1.3 ms, the frequency was 75 Hz, the cycle filling was 1/10 and the magnetic field intensity was 2.3 mT	COVID-19 patients	Melatonin as well as magnetic fields suppress of inflammatory signals ↑	Immunity modulation antioxidant properties and mitochondrial function; antiviral properties suppression in IL-1β and TNF-α	[21,22,28,29,30,31]
6.	Treatment of macrophages with gadolinium and a static magnetic field of high induction	Mice	Gadolinium increased pro-inflammatory (M1) phenotypes, which affected morphology, organelle distribution and gene expression pattern, changing the phenotype of macrophages towards pro-inflammatory and decreased anti-inflammatory (M2) phenotypes of macrophages; magnetic field gradient inhibited pro-inflammatory and increased anti-inflammatory phenotypes of macrophages. ↑ Protective effects of melatonin on macrophages ↑.	Extracellular gadolinium ions block calcium channels, and the force generated by the magnetic field gradient, acting on the cell membrane, changes the localization and/or permeability of calcium channels	[10,51,52]
7.	Shielding of the Earth’s magnetic field HMF can adversely affect neurogenesis, and this effect is closely correlated with ROS concentrations		Magnetic fields of Earth stimulated neurogenesis Melatonin stimulates neurogenesis ↑ HMF ↓ neurogenesis HMF can negatively affect hippocampus-related cognitive functions ↓	Role of ROS in the effects of the hypomagnetic field	[5,24,25,44]
8.	Melatonin and MF treatment of mood disorders	Patients with depression	Melatonin antidepressive action ↑ MF antidepressive action ↑	Effect of MF on serotonin levels Synchronization of diurnal rhythms	[7,38,40,43,50]
9.	Melatonin in glioma MF	Patients with cancers	Complementary therapy Melatonin ↑ MF ↑	Calmodulin-related antiproliferative activity potentially affected Ca^2+^ stimulation apoptosis and decreased angiogenesis and cytoprotective effects by redox status. It suppressed tumor angiogenesis and microcirculation and enhanced the immune response. At the cellular level, it affected tumor cell growth and mitochondrial function and suppressed tumors by interfering with DNA synthesis and reactive oxygen species levels; however, but environmental MFs are cancer promoters	[13,14,16,32,33,34]
10.	Melatonin in neurological disorders such as parkinsonism, Alzheimer’s disease, cerebral edema and traumatic brain injury, depression, cerebral ischemia and amyloidosis	Patients with cognitive disorders and dementia	Improving cognitive functions Melatonin ↑ MF ↑	Antioxidant action detoxifies free radical anti-inflammation mechanisms, immunomodulatory changes, protects neurons and treats neurodegeneration against endoplasmic reticulum stress	[9,13,23,39,41,42,53]
11.	Pancreatitis	Patients with inflammation	Protective and inflammation-reducing effects Melatonina ↑ MF ↑	Antioxidant action detoxifies free radical anti-inflammation mechanisms	[6,32]

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
