# Peer review of "The Physiological Impact of Melatonin, Its Effect on the Course of Diseases and Their Therapy and the Effect of Magnetic Fields on Melatonin Secretion—Potential Common Pathways of Influence"

_biomolecules, 2024, doi:10.3390/biom14080929_

Round 1

Reviewer 1 Report

Comments and Suggestions for Authors

This is well written and informative review important to many research and clinical fields. My only comment is to add the effect and functions of melatonin on immune cells such as macrophages, and also the effect of magnetic forces on the immune cells and the thoughts on how melatonin in the presence or absence of gadolinium based MRI contrast agents could affect the immune system. below are some o examples of citation to be added:

Vuković M, Nosek I, Boban J, Kozić D. Pineal gland volume loss in females with multiple sclerosis. Front Neuroanat. 2024 May 15;18:1386295. doi: 10.3389/fnana.2024.1386295. PMID: 38813079; PMCID: PMC11133707.

Huang Y, Xu Y, Huang Z, Mao J, Hui Y, Rui M, Jiang X, Wu J, Ding Z, Feng Y, Gu Y, Chen L. Melatonin and calcium phosphate crystal-loaded poly(L-lactic acid) porous microspheres reprogram macrophages to improve bone repair. J Mater Chem B. 2024 Jun 28. doi: 10.1039/d3tb02965d. Epub ahead of print. PMID: 38940905.

Xu W, Wu Y, Wang S, Hu S, Wang Y, Zhou W, Chen Y, Li Q, Zhu L, Yang H, Lv X. Melatonin alleviates septic ARDS by inhibiting NCOA4-mediated ferritinophagy in alveolar macrophages. Cell Death Discov. 2024 May 24;10(1):253. doi: 10.1038/s41420-024-01991-8. PMID: 38789436; PMCID: PMC11126704.

Xu MM, Kang JY, Wang QY, Zuo X, Tan YY, Wei YY, Zhang DW, Zhang L, Wu HM, Fei GH. Melatonin improves influenza virus infection-induced acute exacerbation of COPD by suppressing macrophage M1 polarization and apoptosis. Respir Res. 2024 Apr 27;25(1):186. doi: 10.1186/s12931-024-02815-0. PMID: 38678295; PMCID: PMC11056066.

Chanana P, Uosef A, Vaughn N, Suarez-Villagran M, Ghobrial RM, Kloc M, Wosik J. The Effect of Magnetic Field Gradient and Gadolinium-Based MRI Contrast Agent Dotarem on Mouse Macrophages. Cells. 2022 Feb 22;11(5):757. doi: 10.3390/cells11050757. PMID: 35269379; PMCID: PMC8909262.

Fernie KJ, Bird DM. Evidence of oxidative stress in American kestrels exposed to electromagnetic fields. Environ Res. 2001 Jun;86(2):198-207. doi: 10.1006/enrs.2001.4263. PMID: 11437466.

Comments on the Quality of English Language

minor,  grammar needs editing

Author Response

This is well written and informative review important to many research and clinical fields. My only comment is to add the effect and functions of melatonin on immune cells such as macrophages, and also the effect of magnetic forces on the immune cells and the thoughts on how melatonin in the presence or absence of gadolinium-based MRI contrast agents could affect the immune system. Below are some o examples of citation to be added.

Dear Reviewer, 

thank you very much for your insightful comments. The authors will highlight topics related to immune cells and the effects of melatonin in the presence of MRI contrast agents (gadolinium). The authors will make use of some exemplary literature suggestions.

In addition, the English language of the manuscript was improved.

Reviewer 2 Report

Comments and Suggestions for Authors

1.       the title needs to be changed - (patients and healthy people) should be more precise, it does not correspond to the content of the publication

2.       The abstract requires improvement (melatonin is not an archaic compound) it is a biologically active substance, The goal requires change, the abstract is written incomprehensibly and should be corrected

3.       In the introduction chapter, the authors provide very general information (I suggest not to cdefine melatonin a super hormone), it is a hormone, in this chapter I propose to focus on the role of light and melatonin in the function of animals

4.       The chapter The Physiological Role of Melatonin does not concern its role in the body. The authors describe the history of the discovery of melatonin, provide data that have been known for many years, this chapter should be modified and describe the role of this hormone in animals (impact on reproduction, sleep regulation, many others). The information provided does not add anything new

5.       The chapter Influence of magnetic fields on living organisms is described correctly and clearly

6.       In the chapter Potential effects of magnetic fields on melatonin secretion and subsequent possible  health effect, the authors should focus on describing the effect of the magnetic field on melatonin, they describe research on changes in the serotonin profile, the influence of the magnetic field on the melatonin profile is not very clear. The chapter requires changes and ordering of information (some of it is repeated from earlier chapters)

Author Response

Dear Reviewer, 

The authors kindly thank you for the interesting and inspiring review. It allows us to look at our work from different aspects and to present a range of possibilities in looking at the encounter between melatonin and the magnetic field in their millions of years of coexistence.

Response 1.

The title needs to be changed - (patients and healthy people) should be more precise, it does not correspond to the content of the publication.

The authors agree with the caveat about the title; indeed, it does not reflect the content of the paper and the intention to confront many facts about the unusual substance that is melatonin and the magnitude and mode of action of magnetic fields on animate substance along with the interactions possible for them. The title has been changed and is presented as follows: 

The physiological impact of melatonin, its effect on the course of diseases and their therapy, and the effect of magnetic fields on melatonin secretion - potential common pathways of influence.

Response 2.

The abstract requires improvement (melatonin is not an archaic compound) it is a biologically active substance, The goal requires change, the abstract is written incomprehensibly and should be corrected.

The authors thank you for your comments on the abstract. The term archaic they used has to do with the fact that the presence of melatonin is found in artifacts from many millions of years ago, in both plant and animal organisms. The reviewer's attention was taken into account by replacing the description of melatonin. The rest of the abstract was also partially modified, suggesting the reviewer's opinion.

Response 3.

In the introduction chapter, the authors provide very general information (I suggest not to define melatonin a super hormone), it is a hormone, in this chapter I propose to focus on the role of light and melatonin in the function of animals.

The authors thank you for your comment on the introduction, requiring some clarification. The preface, while touching on fairly basic facts, seems necessary in this form, as it serves as an introduction to the subject not only for specialists in the field, but also for other readers wishing to learn more about the subject. In addition, the functions of melatonin are explained in an elaborate table. Focusing on the role of light and melatonin in animals is not the purpose of the article. Examples of melatonin's action under laboratory and environmental conditions in animals are introduced to make it easier to move on to the topics that are the purpose of the article, that is, to find connections between the regulatory function of melatonin and the action of magnetic fields in all their diversity. The naming of melatonin as a superhormone is not intended to classify it in a special way, it is here rather a literary effort to attract the attention of people who have not yet been exposed to this topic and emphasizes the multifaceted function of melatonin.

Response 4.

The chapter The Physiological Role of Melatonin does not concern its role in the body. The authors describe the history of the discovery of melatonin, provide data that have been known for many years, this chapter should be modified and describe the role of this hormone in animals (impact on reproduction, sleep regulation, many others). The information provided does not add anything new.

The authors thank you for drawing attention to some aspects of the description of the physiological action of melatonin. The description of the history of melatonin research takes up only one paragraph and is intended to give an idea of the importance of its discovery, and then leads to the development of the topic in all its complexity. The information contained in this chapter is intended to facilitate the subsequent correlation of the role of melatonin and magnetic fields for living organisms, and to trace possible interaction links for therapeutic and diagnostic benefits. The role of this hormone in animals (impact on reproduction, sleep regulation is described in recently published articles, among others: Korf HW, von Gall C. Mouse Models in Circadian Rhythm and Melatonin Research. J Pineal Res. 2024 Aug;76(5):e12986. doi: 10.1111/jpi.12986. PMID: 38965880, Zhou C, Hu Z, Liu X, Wang Y, Wei S, Liu Z. Disruption of the peripheral biological clock may play a role in sleep deprivation-induced dysregulation of lipid metabolism in both the daytime and nighttime phases. Biochim Biophys Acta Mol Cell Biol Lipids. 2024 Jul 2;1869(7):159530. doi: 10.1016/j.bbalip.2024.159530. epub ahead of print. PMID: 38964437, Zhao B, Yu Z, Sun J, Cheng W, Yu T, Yang Y, Wei Z, Yin Z. Light pollution during pregnancy influences the growth of offspring in rats. Ecotoxicol Environ Saf. 2024 Jul 1;279:116485. doi: 10.1016/j.ecoenv.2024.116485. epub 2024 May 23. PMID: 38788564 and many others.

Response 5.

The chapter Influence of magnetic fields on living organisms is described correctly and clearly

The authors thank you for your opinion expressed on chapter 4.

Response 6.

In the chapter Potential effects of magnetic fields on melatonin secretion and subsequent possible health effect, the authors should focus on describing the effect of the magnetic field on melatonin, they describe research on changes in the serotonin profile, the influence of the magnetic field on the melatonin profile is not very clear. The chapter requires changes and ordering of information (some of it is repeated from earlier chapters).

The authors thank you for your feedback on the chapter on the effects of magnetic fields on melatonin secretion. Melatonin production and secretion is directly related to the rate of conversion of serotonin to melatonin. This topic undertaken in Chapter 5 is only a few-sentence paragraph and indicates that there is no effect of magnetic fields on this transformation and is relevant because of the possibility of increased SAD (seasonal affective disorder) after magnetostimulation. Some of the information regarding melatonin concentrations is repeated intentionally, but in a different context than before, so that certain relationships related to the response of living organisms to magnetic fields can be better seen. Importantly, as the authors repeatedly emphasize in this and other works, there is no concept of a uniform magnetic field, but an infinite number of combinations of its parameters. Therefore, it is always important to underline to which combination of field parameters the organism's reaction occurred, and it is known that it depends on the applied parameters.

Reviewer 3 Report

Comments and Suggestions for Authors

Dear Authors, 

you made a great work! 

However, some improvements are suggested before publication. 

The paper is a review on the effect of magnetic fields on diurnal cycle melatonin secretion in patient and healthy subjects.

The Authors made a great work in terms of methodology and the paper sounds scientific and well written.

However, some improvements are mandatory before acceptance.

I think the title could be written better in English style.

The abstract is well written, complete and summary in its various aspects. The keywords are complete and appropriate.

In the introduction:

·       I think the introduction is absolutely well written and full of interesting ideas and considerations. I suggest the authors revise the English.

The section “2. The physiological role of melatonin” is really well written and complete. I think it could be streamlined in some places, but overall it's well written.

The section “3. Influence of melatonin on the course of diseases and their therapy” is really well written. In this section “These mechanisms are responsible for the increased incidence of hormone-depend-193 ent cancers and perhaps other conditions such as mental, cardiovascular, diabetes, the so-194 called diseases of civilization.” I suggest adding a reference about this topic.

The section “4. Influence of magnetic fields on living organisms” it is well developed. I suggest the Authors review some typos present in the text, and increase the bibliography regarding this extremely interesting topic.

The section “5. Potential effects of magnetic fields on melatonin secretion and subsequent possible health effects” is the main topic of the manuscript, and this aspect is well underlined.

I believe that overall the review is narrative, complete but with the limits of a review of this type, where the manuscripts are selected at the discretion of the Authors, and the conclusions are therefore of little scientific evidence. I therefore suggest that the authors underline this aspect in the manuscript, as limitations of the study.

Conclusions are concise and clear.
Bibliography should be formatted respecting the journal’s requirements and no improper citations are evidenced.

Comments on the Quality of English Language

English is clear and easy to understand.

Author Response

Dear Authors, 

you made a great work! 

Dear Reviewer,

The authors thank you for your supportive and open approach to the reviewed work.

Response 1.

However, some improvements are suggested before publication. 

The paper is a review on the effect of magnetic fields on diurnal cycle melatonin secretion in patient and healthy subjects.

The Authors made a great work in terms of methodology and the paper sounds scientific and well written.

However, some improvements are mandatory before acceptance.

I think the title could be written better in English style.

Dear reviewer, thank you once again for your favor. The authors agree with the fact that the title is imperfect, especially since it does not fully reflect the topic and purpose of the work. The title has been changed and reads:

The physiological impact of melatonin, its effect on the course of diseases and their therapy, and the effect of magnetic fields on melatonin secretion - potential common pathways of influence.

Response 2.

The abstract is well written, complete and summary in its various aspects. The keywords are complete and appropriate. 

Thank you for your positive opinion in this regard.

Response 3.

In the introduction:

 I think the introduction is absolutely well written and full of interesting ideas and considerations. I suggest the authors revise the English.

The authors thank you for the suggestions expressed, some corrections have been made to the language

Response 4.

The section “2. The physiological role of melatonin” is really well written and complete. I think it could be streamlined in some places, but overall it's well written.

The authors thank you for your opinion in this matter

Response 5.

The section “3. Influence of melatonin on the course of diseases and their therapy” is really well written. In this section “These mechanisms are responsible for the increased incidence of hormone-depend-193 ent cancers and perhaps other conditions such as mental, cardiovascular, diabetes, the so-194 called diseases of civilization.” I suggest adding a reference about this topic. 

The authors added references for the pathological processes mentioned.

Response 6.

The section “4. Influence of magnetic fields on living organisms” it is well developed. I suggest the Authors review some typos present in the text, and increase the bibliography regarding this extremely interesting topic. 

The authors corrected typos errors occurring in the paper and expanded the literature in the area mentioned.

Response 7.

The section “5. Potential effects of magnetic fields on melatonin secretion and subsequent possible health effects” is the main topic of the manuscript, and this aspect is well underlined.

I believe that overall the review is narrative, complete but with the limits of a review of this type, where the manuscripts are selected at the discretion of the Authors, and the conclusions are therefore of little scientific evidence. I therefore suggest that the authors underline this aspect in the manuscript, as limitations of the study.

The authors added the range of limitations that exist in the work, related in part to its assumptions, which are presented here as hypotheses and could not yet be proven scientifically. 

Response 8.

Conclusions are concise and clear.

Bibliography should be formatted respecting the journal’s requirements and no improper citations are evidenced.

The bibliography has been revised in accordance with the rules specific to the journal Biomolecules.

In addition, the English language of the manuscript was improved.

Round 2

Reviewer 3 Report

Comments and Suggestions for Authors

Dear Authors, 

I consider the manuscript suitable for publication. 

Author Response

Dear Authors, 

I consider the manuscript suitable for publication. 

Dear Reviewer,
Thank you very much for helping us improve the manuscript.

Best regards,